# Diet in the Treatment of Epilepsy: What We Know So Far

**DOI:** 10.3390/nu12092645

**Published:** 2020-08-30

**Authors:** Alberto Verrotti, Giulia Iapadre, Ludovica Di Francesco, Luca Zagaroli, Giovanni Farello

**Affiliations:** Department of Pediatrics, University of L’Aquila, Via Vetoio, 1. Coppito, 67100 L’Aquila, Italy; giuliah@live.it (G.I.); difrancesco.ludovica@gmail.com (L.D.F.); lucazaga@hotmail.com (L.Z.); giovanni.farello@cc.univaq.it (G.F.)

**Keywords:** Atkins diet, branched chain amino acids (BCAA), caloric restriction, epilepsy, gluten, herbal remedies

## Abstract

Epilepsy is a chronic and debilitating neurological disorder, with a worldwide prevalence of 0.5–1% and a lifetime incidence of 1–3%. An estimated 30% of epileptic patients continue to experience seizures throughout life, despite adequate drug therapy or surgery, with a major impact on society and global health. In recent decades, dietary regimens have been used effectively in the treatment of drug-resistant epilepsy, following the path of a non-pharmacological approach. The ketogenic diet and its variants (e.g., the modified Atkins diet) have an established role in contrasting epileptogenesis through the production of a series of cascading events induced by physiological ketosis. Other dietary regimens, such as caloric restriction and a gluten free diet, can also exert beneficial effects on neuroprotection and, therefore, on refractory epilepsy. The purpose of this review was to analyze the evidence from the literature about the possible efficacy of different dietary regimens on epilepsy, focusing on the underlying pathophysiological mechanisms, safety, and tolerability both in pediatric and adult population. We believe that a better knowledge of the cellular and molecular biochemical processes behind the anticonvulsant effects of alimentary therapies may lead to the development of personalized dietary intervention protocols.

## 1. Introduction

Epilepsy is one of the most common neurological diseases, with a worldwide prevalence of 0.5–1% and a lifetime incidence of 1–3% [1]. This chronic condition is often controlled through a pharmacological approach; however, 30% of affected patients develop a drug–resistant epilepsy (DRE), which is defined after the “failure of adequate trials of two tolerated and appropriately chosen and used antiepileptic drugs schedules (whether as monotherapies or in combination) to achieve sustained seizure freedom” [2]. Patients with DRE should resort to epilepsy surgery as last treatment option, after trying alternative non-drug treatments, such as vagus nerve stimulation and restrictive diets [3]. The classic ketogenic diet (KD) has long been used in the treatment of refractory epilepsy in children and adults or in patients not candidates for surgery, showing evidence of efficacy especially in the pediatric population, although KD remains a valid therapeutic choice [4,5]. In the last 20 years, in addition to the classic form of KD, new variants have been introduced, including the modified Atkins diet (MAD), the low-glycemic index treatment (LGIT), and the medium-chain triglyceride diet (MCTD) for the treatment of epilepsy and other neurological diseases [6]. The main types of KD are represented in Table 1.

All have shown efficacy against refractory epilepsy, maintaining a good safety and tolerability profile. In this paper, we reviewed clinical efficacy of KD in children and adult epilepsy, safety, and tolerability, in addition to provide special consideration when treating different age populations.

## 2. Different Dietary Regimens Inducing Ketosis

### 2.1. Classic Ketogenic Diet (KD)

The classic KD was first introduced in 1921 by Wilder for treatment of epilepsy to avoid malnutrition that occurs with prolonged fasting [10]. KD has a high fat content (80–90%), and low protein (6–8%) and carbohydrates (2–4%), with the aim of inducing ketosis. KD mimes a fasting situation, allowing for a limited intake of fluids and calories. In this condition, the cellular metabolism uses fatty acids as the primary source of energy, while the hepatic catabolism determines the formation of ketone bodies with consequent urinary ketosis [11]. The fat per protein plus carbohydrates ratio is 4:1, i.e., four portions of fatty acids and one of proteins and carbohydrates must be introduced with the diet. In children who need more protein for growth, this ratio can be changed to 3.5:1 or 3:1. To achieve this, two different approaches can be used. In the first method, the fasting patient is hospitalized for 12–48 h, or when ketones are present in the urine to monitor the state of hydration and blood sugar [12]. With this approach, the patient reaches ketosis faster but at the cost of greater stress [13]. Once ketosis is achieved, meals are calculated to maintain a constant KD ratio. In the second method, hospitalization is not required and the KD ratio is gradually increased weekly from 1:1 until 4:1 [4]. According to the current literature, there is no significant difference between the two diet approaches in the time required to achieve ketosis; therefore, it is preferable to use the second method, which does not require fasting. Due to the reduced intake of fruit, vegetables, milk and derivatives, their integration is necessary. Multivitamin complexes and mineral supplements should be taken daily [3].

### 2.2. Modified Atkins Diet (MAD)

The MAD has been used since 2003 to treat children and adults with refractory epilepsy at Johns Hopkins Hospital, in Maryland (USA) [14,15]. It has a high fat content (65%) and a low protein (25%) and carbohydrate (10%). The fat per protein and carbohydrate ratio is 1:1, representing a more palatable and less restrictive variant of KD, suitable for children or patients with behavioral problems [7]. There is no limitation in the intake of liquids, calories, and/or proteins but there is a restriction in carbohydrates, with a maximum intake of 10–20 g/day in children and 15–20 g/day in adults [16,17]. Reduced glucose intake can cause urinary ketosis [18]. Daily intake of multivitamin complexes and calcium carbonate supplements is recommended. The diet can be initiated in an outpatient setting, fasting and weighing of foods are not required [17].

### 2.3. Low-Glycemic Index Treatment (LGIT)

The LGIT was first used for refractory epilepsy treatment in 2005 at Massachusetts General Hospital, Boston (USA) [19]. It is a less restrictive diet, which has a high content of fat (60%), an amount of protein greater than other diets (20–30%) and a low carbohydrate content (10%), with a low glycemic index (GI) [7]. In this diet, meals with a glycemic index of less than 50, such as meat, dairy products, some fruits, whole grains, and bread are allowed [8]. The fat per protein and carbohydrate ratio is 1:0.6. There are no limitations in the introduction of liquids and calories [20]. Compared to KD, LGIT produces fewer ketone bodies, but it seems to be better tolerated [19,21].

### 2.4. Medium-Chain Triglyceride Diet (MCTD)

The MCTD was first introduced by Huttenlocher et al. [22]. In 2008, it was modified by Neal et al. [23]. It is a very flexible diet, with a high fat content (30–60%), and low protein (10%) and carbohydrates (15–19%). Medium-chain triglyceride (MCT) fat produces more ketones per gram than long-chain triglyceride (LCT) fat, used in the classic KD [9]. The high ketogenic potential allows to reduce the intake of fatty acids in favor of a greater consumption of proteins and carbohydrates, making the diet more palatable and usable for children compared to KD [8]. Moreover, children on MCTD tend to grow better, require fewer micronutrients, and have a significantly lower total cholesterol/high density lipoprotein ratios compared to the classic KD [24].

## 3. Responders and Non-Responders to the Ketogenic Diet

In clinical practice, there are KD responders, who have a ≥50% seizure-frequency reduction, and non-responders [25]. Specific metabolic alterations, such as type 1 glucose transporter deficiency syndrome (GLUT1) and pyruvate dehydrogenase complex deficiency, have been reported to be associated with a favorable response to KD [26]. Studies have shown that the efficacy of KD differs based on specific genetic mutations in patients with developmental and epileptic encephalopathy (DEE). Responders are patients with DEE and mutations, such as SCN1A, KCNQ2, STXBP1, or SCN2A, while non-responders are those with CDKL5 mutations. Therefore, in this particular group of subjects, the identification of the pathological mutation can facilitate the clinician in predicting the efficacy of the KD response [27]. Although larger cohort studies are needed, KD is less effective in patients with the minor allele of rs12204701 and mutation in the CDYL gene [26]. Unfortunately, other predictors of the KD response are currently unknown. It is important to know why some individuals respond to the diet and others do not, in order to develop more appropriate dietary approaches [25].

## 4. Pathophysiology of Ketogenic Diet and Epilepsy

The mechanisms involved in KD-induced seizure reduction have not yet been fully understood; several hypotheses and theories have been formulated. Potential pathophysiological anti-seizure mechanisms of KD are shown in Table 2 and Figure 1.

It has been seen that the ketone bodies (KB) and polyunsaturated fatty acids (PUFAs), which can be enhanced under KD, may play main roles in producing anti-seizure effects [8]. During KD, energy is produced through the fatty acids oxidation in the mitochondria with a consequent increase in the production of acetyl-CoA. KB, such as acetoacetate and beta hydroxy butyrate, are synthesized from acetyl-CoA mainly in the liver, and are then introduced into the bloodstream. KB are used as an alternative source of energy instead of brain glucose in individuals on KD. When the KB arrive in the brain, they are converted into acetyl-CoA and enter the cycle of tricarboxylic acid at the level of the brain mitochondria, by which adenosine triphosphate (ATP) is obtained [32].

Possible antiepileptic mechanisms of action of KB include changes in neurotransmitters and polarity of the neuronal membrane, alterations in neuronal energy metabolism, effects on neuronal membrane ion channels, and neuroprotective action against oxidative stress [31].

According to the current literature, patients taking KD usually show an increase in inhibitory neurotransmitters in the brain, such as gamma-aminobutyric acid (GABA), agmatine, monoamines, galanin, and neuropeptide Y, with a decrease in the excitability of the neuronal membrane and an augmentation of the seizure threshold [33]. Indeed, some studies show an increase in the concentration of gamma-aminobutyric acid (GABA), the main CNS inhibitory neurotransmitter, in the cerebrospinal fluid (CSF) of patients with KD [34]. Furthermore, during KD, there is a reduction of aspartate, which is an inhibitor of glutamate decarboxylase; this enzyme normally catalyzes the α-ketoglutarate, which is a precursor of GABA, with a consequent increase in the synthesis of GABA [35]. It has been hypothesized that the low levels of aspartate induced by ketosis would support the excitatory glutamate conversion into glutamine in astrocytes. In turn, glutamine would be incorporated by neurons and converted into GABA, which would perform its inhibitory functions [36]. High levels of agmatine, a small inhibitory neurotransmitter with neuroprotective properties, have been found in the hippocampus in animal models treated with KD [32]. Monoamine neurotransmitters, including norepinephrine, dopamine, and serotonin, also have ongoing KD changes. Adenosine A1 is increased like GABA, suggesting greater activation of the noradrenergic nervous system and its anticonvulsant effect in KD. It seems to act with a combined mechanism, both reducing neuronal excitation through presynaptic inhibition and inducing neuronal hyperpolarization by postsynaptic activation potassium channels [37]. In a study involving DRE children, a change in serotonin and dopamine levels in the CSF attributable to KD was observed [38]. In addition, during KD a fasting condition is mimicked with a reduction of glucose levels but also of insulin and leptin, with consequent overexpression of galanin and neuropeptide Y. The latter have an important anticonvulsant effect thanks to the inhibition of neuronal excitability [39,40,41].

During KD, there is an enhancement of brain energy production. In the beginning, this may expose pre-existing mitochondria to a greater workload, and consequently to a greater oxidative stress. Subsequently, the KD determines an over-expression of the genes of the energy metabolism, an increase in the mitochondrial number and biogenesis and rise in glutathione to deal with reactive oxygen species (ROS) [42].

Overall, greater energy production in the form of ATP and reduction of oxidative stress are obtained, with an increase in energy reserve in the form of phosphocreatine, improvement in neuronal homeostasis and greater resilience in stressful conditions [43,44].

ATP-sensitive potassium (KATP) channels seem to play a key role in the anticonvulsant effects of KD [45]. During KD, the reduction of brain glucose and the changes in the ATP/ADP ratio induces a greater opening of the KATP channels at the level of the neuronal membranes with subsequent hyperpolarization, reduction of neuronal excitability and increase in the seizure threshold [28,46]. According to some studies, KB would increase the activation of two-pore domain potassium (K2P) channels, allowing a continuous outflow of K ions [47,48]. Furthermore, in the course of KD there is an increase in the synthesis of some species of fats, in particular of the PUFAs, which play an important role for their anticonvulsant effect [49,50,51,52]. During KD, PUFAs bind and activate the peroxisome proliferator-activated receptor gamma (PPARγ) isoform, a transcription factor that regulates anti-inflammatory, antioxidant, and mitochondrial genes inducing greater stability of synaptic function, reduction of neuronal excitability and increase in the energy reserve [28]. In addition, PUFAs modulate the activity of ion channels, i.e., activate K2P channels and the Na/K ATPase and block voltage-gated sodium and calcium channels, increasing the seizure activation threshold and reducing neuronal excitability [28]. PUFAs also exert anticonvulsant effects indirectly, by increasing the expression of uncoupling proteins (UCP) or reducing the production of ROS [28,53].

Over time, seizures are harmful since they can cause cell damage and even neuronal death [8,54]. KD appears to protect cells from apoptosis and cell death through increased expression of calbindin, which binds to intracellular calcium, inhibition of various pro-apoptotic factors, such as caspase 3, and inhibition of pore formation in mitochondria [29,55,56,57].

The most recent hypothesis is that changes in the intestinal microbiota, induced by KD, have repercussions on seizures. Olson et al. highlighted how KD modifies the intestinal microbiota, resulting in a reduction in α-diversity and an increase in some positive bacteria Akkermansia muciniphila and Parabacteroides species. Overall, this results in a decrease in gamma-glutamyl amino acids in the blood and an increase in GABA/glutamate in the brain [30]. KD also appears to modify the production of pro-inflammatory cytokines. In mouse models, a reduction in peripheral and brain production of interleukin1b was registered [58].

Although all the mechanisms involved in the reducing seizure induced by KD have not yet been fully elucidated, it has been shown that the ketone bodies and PUFAs, which can be enhanced under KD, may play key roles in their anti-seizure effect. By increasing inhibitory neurotransmitters, activating of potassium channels, and enhancing the energy production of the nervous system, the KB can enhance the brain seizure threshold. On the other hand, fatty acids and PUFAs, particularly n–3 PUFAs, have been revealed to activate peroxisome proliferator-activated receptors (PPARs), in particular PPARγ, leading to up-regulation of energy transcripts, enhancement of energy reserves, and stabilization of synaptic function that eventually prevents neuronal hyperexcitability. PUFAs can also alter ion channels activities that cause neuronal hyperpolarization.

Furthermore, upregulation of UCPs induced by PUFAs can diminish oxidative stress. In addition to these mentioned mechanisms, KD may alleviate the seizure-induced neuronal damage through several probable anti-apoptosis and anti-necrosis mechanisms. Overall, KD and its variants not only can eventually limit the seizures, but also may alleviate their adverse effects on neurons through different mechanisms.

## 5. Clinical Efficacy against Epilepsy

Since the introduction of KD as treatment option for refractory epilepsy in 1921, this dietary regimen has been widely employed in patients with DRE. From the end of the last century, its efficacy against epilepsy has been systematically assessed through several studies, both in pediatric and adult populations. Most relevant articles on the use of the KD in pediatric refractory epilepsy and in infantile spasms have been reported in Table 3; Table 4; these articles have been published from 1998 to 2020.

Over the years, observational studies, randomized controlled trials, and meta-analysis have shown that different KDs reduce seizure frequency significantly in children and adolescents with DRE. There has been a gradual transition from a more restrictive KD ratio, i.e., that of the classic KD equal to 4:1, to a less restrictive one, such as that of MCTD or MAD equal to approximately 1:1, with no reduction in efficacy (Table 3). According to current scientific evidence, the KD efficacy for the management of intractable epilepsy would be greater in the pediatric population than in the adult population. However, the studies on adults provide dubious results, are fewer and all conducted on small samples and, therefore, require further investigation [5].

### 5.1. Pediatric Population

#### 5.1.1. Refractory Epilepsy

Evidence of KD efficacy against intractable seizures in pediatrics came from observational, randomized control trials (RCTs) and meta-analysis. In a 2007 retrospective long-term observational study involving children with refractory epilepsy, of the 38 patients (55%) who remained on KD at 1 year, 70% had >75% seizure frequency reduction (SFR), and 25% had 50‒75% SFR, with the better results/greater efficacy observed for generalized epilepsies [61]. In 1998, a multicenter observational study including 51 children with DRE was carried out at tot different. At one year, approximately half (47%) of the children were still on KD and, of these, 43% and 39% experienced >90% and 50‒90% SFR, respectively [59]. Subsequently, two other studies confirmed these positive results [60,62].

The first RCT assessing the efficacy of KD in children was conducted in 2008. It involved 145 pediatric patients aged 2–16 years with DRE, who were randomized to receive KD immediately (KD group) or after three months, while continuing on AEDs at stable doses (control group). At three months, KD group experienced a 75% reduction in seizure frequency compared to the control group; additionally, 38% of patients on KD had >50% SFR and 7% had >90% SFR, showing clear benefits for KD compared to no change in anticonvulsant treatment [23]. In 2016, two different RCTs took place. The first involved 40 pediatric patients aged 12–36 months with symptomatic refractory epilepsy, who were randomly divided to receive KD, to receive MAD or to continue on AEDs. At 6 months follow-up, a SFR >50% was reported for all patients on KD, showing the superiority of ketogenic liquid formula over MAD or anticonvulsant medications [63]. The second included 102 participants aged 2–14 with DRE and daily seizures who had not responded to three AEDs. Patients were randomly assigned to receive MAD or to continue on AEDs over a 3-month period, reporting a >90% SFR in 30% of patients of diet group vs. 7.7% of control group and >50% SFR in 52% of patients of diet group vs. 11.5% of control group (*p* < 0.001) [64]. In 2016, another RCT included 81 children aged 2–14 years with refractory epilepsy and daily seizures who had not responded to at least two AEDs; these were randomized to receive either the simplified MAD or no dietary intervention over 3-month period. The proportion of patients with >50% SFR was significantly greater in the diet group compared to the control group (56.1% vs. 7.5%) [65]. In 2017, two additional RCTs were carried out on KD efficacy against intractable epilepsy in children: a large percentage of patients on KD reported a reduction in severity of their worst seizure compared to control group [67].

A recent meta-analysis reviewed all evidence regarding the efficacy and tolerability of KD and MAD from RCTs in pediatric population with intractable epilepsy. The authors identified five RCTs for a total of 472 patients recruited. This meta-analysis confirmed the efficacy of the KD [68]. Another recent meta-analysis comparing the short-term and long-term efficacy of classic KD and MAD in children and adolescents with epilepsy, revealed no substantial difference between the two dietary regimens [82].

#### 5.1.2. Other Childhood Epilepsies

In addition to refractory epilepsy, there are specific epileptic disorders in which KD has been employed successfully and, therefore, its use may be considered early in the treatment course. These conditions include Dravet Syndrome (DS), myoclonic-astatic epilepsy, infantile spasms (IS), and Febrile Infection Related Epilepsy Syndrome (FIRES).

(1) Dravet syndrome. DS is a very severe myoclonic epilepsy in infancy. It usually emerges in the first year of life with febrile convulsions followed by afebrile generalized seizures in the next years. The seizures are very difficult to control and the infants develop intellectual disability. Twenty-one studies on KD and DS have been published from 2005 to 2020. In DS, evidence about the effectiveness of KD on controlling seizures comes mainly from retrospective observational studies [83,84,85,86]. The diet appears to be particularly effective when used in combination with the gold standard triple therapy of valproic acid (VPA), stiripentol (STI), and clobazam (CLB), in addition to exhibit neuroprotective effects and to control long-lasting SE refractory to conventional AED treatment or prolonged generalized seizures [87,88,89,90]. Basing on animal models, KD might also provide protection from sudden unexpected death in epilepsy (SUDEP) which represents a major cause of mortality in DS patients [91].

(2) Doose Syndrome. Doose syndrome is uncommon epileptic disorder characterized by myoclonus, generalized epilepsy, and neurological deterioration that can have infancy onset. In the literature, there are 17 studies on KD and Doose syndrome, published from 1998 to 2019. KD is the most frequently reported therapy for Doose syndrome and, probably, the most efficacious one, as resulted from retrospective observational studies in which the dietary regimen led to a 50−99% SFR in more than one third of patients and seizure freedom in 18–56%, after multiple other therapies had been tried and failed [92,93,94]. Given the evidence of the benefits of the diet for the treatment of Doose syndrome, in the 2009 the expert consensus guideline for optimal management of KD listed Doose syndrome as one of the eight probable indications for the diet intervention [17].

(3) Infantile spasms. IS is an epileptic encephalopathy which generally occurs in children younger than 1 year; it is characterized by clinical spasms, hypsarrhythmia found in electroencephalogram; frequently, epileptic spasms are accompanied by developmental delay. The use of KD, in this type of epilepsy, is one of the best assessed in many studies, in fact 59 articles have been published from 1975 to 2020. For a long time, KD was not recommended in infancy (under the age of 2 years) because of the specific nutritional requirements typical of this period of life. Anyway, KD is highly effective and well tolerated in infants with epilepsy, with a high rate of seizure freedom ad optimal adherence to the diet therapy. Over the last decade, KD has gained popularity as treatment option for IS. Its efficacy as adjunctive therapy for patients with IS was investigated in a recent systematic review including nine retrospective and four prospective trials. The authors found a spasm reduction >50% in a median rate of 64.7% of patients (IQR: 38.94%), and a median spasm-free rate of 34.61% (IQR: 37.94%), suggesting a potential benefit of KD for drug-resistant IS patients [95]. In a recent RCT, a total of 32 infants with IS were randomized to receive KD or high-dose adrenocorticotropic hormone (ACTH) treatment. KD resulted to be as effective as ACTH in the long term but better tolerated. In patients without prior VGB treatment, ACTH was more effective in achieving short-term remission. However, with prior VGB, KD was equally effective as ACTH in the short term [96]. These findings may lead to consider KD at an earlier stage in the management of IS.

(4) Febrile infection related epilepsy syndrome. FIRES is a rare catastrophic epileptic encephalopathy, characterized by acute onset of recurrent seizures or refractory status epilepticus (SE) preceding febrile illness, in absence of infectious encephalitis; often mental retardation is present. According to some anecdotal reports, i.e., 11 studies published from 2010 to 2020, KD has also shown some efficacy in FIRES [97,98,99]. FIRES is associate with a bad prognosis since available treatments are poorly effective. KD, thanks to its anti-inflammatory properties, may be of great impact in the treatment of FIRES, not only during the acute phase but also in long-term epilepsy management [100]. Moreover, KD seems to exert a positive effect on cognitive outcome and should, therefore, be considered precociously in the course of treatment, as first-line agent [101]. In a recent retrospective observational study, seven patients with FIRES and super-refractory status epilepticus (SRSE) were put on KD. Within 5 days from KD initiation, all patients experienced SE cessation, reduction in the number of seizures per day, and shortening of seizure duration [102].

(5) Myoclonic status in non-progressive encephalopathy. Myoclonic status in non-progressive encephalopathy (MSNPE) has the following features: recurrence of long-lasting atypical status epilepticus associated with attention impairment, continuous polymorphous jerks, and/or other complex abnormal movements; the patients show a non-progressive encephalopathy. KD may have possible beneficial effects also in the treatment of MSNPE, as emerged from a small retrospective study. Six out of 99 patients with MSNPE were placed on KD and followed up for at least 24 months, with most of them (5/6) showing >50% SFR and disappearance of myoclonic SE within 6 months from diet initiation [103].

### 5.2. Adult Population

In recent years, interest in KD as valuable treatment option for refractory epilepsy has spread from pediatrics to adults; this patient population may also benefit from KD, although only a few trials have been published so far. A 2017 meta-analysis included 16 open-label prospective studies for a total of 338 patients aged 16–86 years with DRE treated with KD and its variants. The analysis provided good evidence of efficacy, with 13% of subjects becoming seizure free and 53% showing >50% SFR; in addition, the dietary treatment was found to be well-tolerated and to have acceptable side effects, despite some limitations of the study [104]. For example, the fact that only patients who completed the trials (probably those who had a better response) were included in the analysis; the small sample size, and the presence of several methodological differences among the trials considered. Further research is required to investigate the therapeutic effects of KD in adulthood, focusing on different seizure types, on which type of diet or ratio is more effective in epilepsy treatment.

## 6. Tolerability

The use of KD in drug resistant childhood epilepsy brings many benefits in terms of seizure control and reduction of the number of AEDs; however, this dietary therapy is not as safe as the ordinary diet, potentially leading to different adverse events (AEs). On the other hand, KD seems not to have any negative effect on cognition or behavior. All these considerations should be made when choosing the most appropriate antiepileptic treatment to be proposed to pediatric patients.

In particular, classical KD is often associated with short-term AEs in both the induction and maintenance phases, although they are generally well tolerated and do not require treatment discontinuation. Unfortunately, the ingestion of solid foods versus liquid formula and the restrictive KD ratio appear to play a key role in reducing therapeutic adherence [60,88]. As evidenced by a recent Cochrane review, based on the results of randomized or quasi-randomized clinical trials comparing the efficacy of different KD variant, the classic KD has an efficacy slightly higher than MAD, but it is also less tolerated due to the high number of AEs [5]. MAD is a more palatable and flexible diet that is associated a more liberal KD ratio and fewer AEs than the classic KD and, therefore, it may be the first dietary choice, especially in children with DRE [105]. MCTD and LGIT are also less restrictive and more palatable diets, however further investigations are necessary to know their safety and tolerability. 

### 6.1. General Comment

More than 40 categories of AEs have been described in association with the KD, as emerged from a systematic review on safety and tolerability of KD for the treatment of childhood DRE including 45 studies (of which 7 RCTs) [93]. We have subdivided AEs in short and long term AEs. 

### 6.2. Short Term Adverse Events

The most common AEs related to diet consumption were gastrointestinal disturbances (40.6%), including constipation, vomiting, diarrhea, hunger, abdominal pain, gastroesophageal reflux, and fatty diarrhea. Severe AEs, such as respiratory failure, thrombocytopenic purpura, and pancreatitis, were not frequent, occurring in no more than 0.5% of treated children [106].

### 6.3. Long Term Adverse Events

Among the long term AEs, we must remember the AEs on renal system: a total of 112 patients starting the KD, six reported kidney stones (three uric acid, and three mixed calcium oxalate and uric acid stones) during a follow-up period of 2 months−2.5 years [107]. In fact, patients maintained on KD often have hypercalciuria, acid urine, and low urinary citrate excretion; these, in conjunction with low fluid intake, can lead to a higher risk for ureteral stone formation.

The effect of KD on bone mineral content was assessed by using dual energy X-ray absorptiometry in 25 children who received KD for 15 months; despite supplementation with vitamin D and calcium, a significant decline in both whole-body and spine BMC for age, and in both whole-body and spine BMC for height, were registered. Body mass index (BMI) Z score, age, and ambulation were found to be positive predictors of BMC, suggesting worse BMC status for younger children and not ambulatory subjects or with a low BMI [108].

The long-term influence of KD on blood lipid levels was specifically investigated in two studies. In one, a significant increase in the plasma cholesterol, low-density lipoprotein, very low-density lipoprotein, triglyceride, and total apolipoprotein B levels, and a marked decrease in the mean high-density lipoprotein (HDL) level were reported at 6–24 months of treatment with high-fat KD [109]. In the other trial, KD therapy led to higher percentages of patients with hyperlipidemia and low HDL [110]. In addition, the use of solid food was more likely to induce hypercholesterolemia with respect to the formula-based KD.

As far as nutrient intake is concerned, both KD and MCT diet provide inadequate amounts of most micronutrients without the addition of vitamin and mineral supplements. In particular, mean plasma selenium and magnesium levels decrease in patients on dietary regimen, while phosphorus, and folate did not meet the dietary reference intakes [111,112].

Vascular function seems to be not affected by KD, as demonstrated by two studies through the measurement of the carotid intima-media thickness, the carotid stiffness index, and elastic properties of the aorta in children taking dietary treatment [113,114]. On the other hand, right ventricular diastolic dysfunction may be associated with the use of KD in the short term [115].

The impact of the KD on growth in children was evaluated in several studies. However, results were inconsistent and controversial, due to the variation in population characteristics, diet prescriptions, measured parameters and, mostly to different observation periods among the studies; overall, interval >12 months were associated with a negative influence of KD on growth [106].

### 6.4. Deaths, Retention Rates, and Reasons for Diet Discontinuation

Reported deaths during KD treatment were due to aspiration pneumonia, diabetes mellitus, infection diseases, arrhythmia, shock, drowning, nocturnal seizure, asphyxia related to sputum blockage, perforated gastric ulcer after epilepsy surgery, status epilepticus, and accidental injuries. Anyway, no death was directly attributed to the dietary regimen.

Retention rates for KD at 1 year and 2 years were approximately one half and one third, respectively, while about 10% of children were still on the diet at 3 or 4 years. The main reason for diet discontinuation was the lack of efficacy; while only 11% of the patients dropped out because the diet was too restrictive. More than one-half of subjects discontinued the dietary treatment for either of these two reasons. Side effects were responsible for KD discontinuation in less than 10% of cases and the occurrence of AEs was even rarer in prospective studies of longer duration [106]. In the current literature, the mortality rate associated with KD use remains unknown, as this information is based on case reports or case series.

## 7. Special Considerations

### 7.1. Exclusion Criteria of the Ketogenic Diet 

KD offers a treatment option for intractable epilepsy with confirmed efficacy in both pediatric and adult populations. Anyway, its use may not be appropriate for certain categories of patients and, therefore, the suitability of the diet should be carefully assessed before its prescription. Fatty acid oxidation defects, carnitine deficiencies, organic acidurias, pyruvate carboxylase deficiency, familial hyperlipidemia, hypoglycemia, ketogenesis/ketolysis defects, severe gastroesophageal reflux, severe liver diseases, and disorders needing a high carbohydrate diet such as acute intermittent porphyria represent definite contraindications and must be excluded prior to considering KD initiation. 

Diabetes mellitus, certain mitochondrial diseases, and concomitant steroid use may also constitute possible contraindications to the dietary intervention. The presence of clinical features such as developmental delay, cardiomyopathy, hypotonia, exercise intolerance, myoglobinuria, and easy fatigability suggest that the child should be tested to rule out an inborn error of metabolism. Thus, a careful evaluation of the patients including neurological examination should always be performed prior to KD administration [8]. As for obese patients, KD helps to lose weight in the short term with possible improvement in blood glucose, insulin, and blood pressure values [116]. However, the lack of long-term studies and the complications associated with KD, such as hypoglycemia, dehydration and electrolyte disturbances, make it impossible to recommend its use in the prevention of cardiovascular disease [116,117]. Further studies on sustainability, safety, efficacy, duration of KD, prognosis after treatment discontinuation should be performed.

### 7.2. Variation of Ketogenic Diet Regimen per Age Group

KD is considered to be an effective and well-tolerated treatment option in all pediatric ages from infancy to adolescence, until adulthood. However, many KD variants exist, which differ to each other for macronutrients composition, fat to carbohydrate plus protein ratio, palatability, and management of the diet. All these characteristics make a diet more or less suitable in the various stages of life. There is evidence that a rigid and strict initiation protocol with carbohydrate restriction and increase fat intake is associated with a slightly higher efficacy; anyway, other factors should be considered in designing a diet, rather than efficacy, such as the patient condition, severity sand type of epilepsy, in addition to family environment [31]. In this regard, infancy and neonatal age represent a target population as “fragile” as promising for the use of KD. In a 10-year, retrospective, observational study involving 115 infants treated with KD (70%) or MAD (30%), the authors found a high rate of long-term seizure-free outcomes (37% of the patients at 12 months), which can be predicted based on the seizure freedom at three months (50% of the patients) regardless of etiology (genetic, structural brain abnormalities, and metabolic pathologies) [118]. In fact, the higher amounts of enzymes that metabolize ketones and the production of monocarboxylic acid transporters, leading to more ketone bodies that can cross the blood-brain barrier, are responsible for the greater efficacy of such dietary treatment in infants. In the first year of life, ketones actively participate to myelin formation. Furthermore, the availability of several ketogenic formulas has resulted in an increase compliance with the dietary treatment, contributing to a wide spread of KD among infants and neonates in the last years. In this age population, the classic type of KD appears to be more suitable and efficient.

Although the classic form is the most efficient, it appears to be also quite restrictive and time-consuming and, therefore, not suitable in any condition. For example, in adults and adolescents, less restrictive and more suitable KD variants, such as MAD and LGIT should be preferred, showing comparable efficacy in older children [3].

For enterally fed infants and children, KD should be given as liquid formula, which is more efficient and easy to administer.

## 8. Conclusions

The KD is an effective, relatively safe, and tolerable dietary treatment for adults and children with refractory epilepsy. In the last decades, less restrictive and more liberal KD variants have been developed to make the diet more feasible and palatable while reducing sides effects and making it available for a larger group of patients. KD has a major benefit compared with standard anticonvulsant treatment with AEDs, which is associated with long-term side effects.

The choice of the most appropriate dietary regimen must be made on an individual basis considering the patient’s age and conditions, family environment, epilepsy type, nutrition status of epileptic patient, and responsiveness of epilepsy to other treatment modalities. Most prominent AEs are related to the GI tract and may be reduced by small adjustments of the diet (e.g., fluid intake adjustments; laxatives); the impact on growth is controversial and its negative influence may be detected over longer periods of time. Thus, long-term follow-up is required to observe whether temporarily altered lipid profile while on the KD may lead to higher risk of developing atherosclerosis or cardiovascular disease later in life and to evaluate long-term growth outcomes. Children on KD should be followed up regularly and monitored for years, even after stopping the diet.

Given the beneficial clinical results regarding efficacy and safety, neurologist should be able to refer appropriate patients to the KD as soon as necessary instead of considering it as a last option only. KD may be used as an adjunct treatment to AEDs in children and adults with refractory epilepsy. However, in some metabolic diseases, such as type 1 glucose transporter and pyruvate dehydrogenase deficiencies, and mitochondrial complex I defect, KD may be considered a first-line of treatment. About the current limitations, the most important is the lack of data about a possible role of KD as a single therapy for management of intractable epilepsy [119]. This dietary therapy should be offered earlier to a child after the failure of two AEDs appropriately used.

Moreover, another gap is that, although KD is a useful treatment in various types of severe epilepsy, we need more information about its possible use in other types [120]. Perhaps, in the next years, we will observe a great increase of data about the physiopathological mechanisms of KD. These data could contribute to understanding what epileptic syndromes can be treated by KD. Moreover, more reliable data obtained from large studies about the efficacy and safety of KD will be available. Finally, the least important gap is the few data about some specific conditions in which the KD may be particularly effective, such as myoclonic epilepsies, including Dravet Syndrome and Doose Syndrome.

In conclusion, KD can also represent a valid option for the treatment of IS. Promising data also come from its use in FIRES and MSNPE, although future, larger RCTs are needed to support the initial findings.

## Figures and Tables

**Figure 1 nutrients-12-02645-f001:**
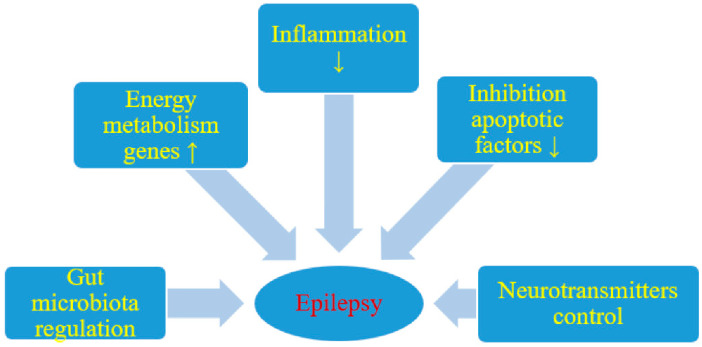
Likely effect of a ketogenic diet on seizure activity; ↑ increase, ↓ decrease. Reproduced with permission from (Ułamek-Kozioł, M.; Czuczwar, S.J.; Januszewski, S.; Pluta, R.), (Nutrients); (2019). Reference: [31].

**Table 1 nutrients-12-02645-t001:** The main types of Ketogenic diets.

Ketogenic Diets	Macronutrients (% Total Daily Calories)			Ketogenic Ratio
	Fat	Protein	Carbohydrate	
Classic	80–90	6–8	2–4	4:1
MAD	65	25	10	1:1
LGIT	60	20–30	10	1:0.6
MCTD	30–60	10	15–19	1:1 or 2:1

Abbreviation: MAD, modified Atkins diet; LGIT, low-glycemic index treatment; MCTD, medium-chain triglyceride diet. References: [7,8,9].

**Table 2 nutrients-12-02645-t002:** Potential pathophysiological anti-seizure mechanisms of Ketogenic Diet.

Primary Physiological Alteration	Possible Anti-Seizure Mechanisms
(1) Increase in Ketone Bodies (KB)	Elevation of inhibitory neurotransmitters and reduction of neuronal excitability. Brain energy production enhancement (through upregulation of energy metabolism genes, elevation of mitochondrial density and biogenesis, increase in energy reserves) with improvement in neuronal homeostasis. Neuronal hyperpolarization by potassium channel activation (ATP-sensitive potassium (KATP) and two-pore domain potassium (K2P) channels) Slow energy production provided by the KB (compared to the energy that comes from glucose) appears to have antiepileptic effects.
(2) Increase in Polyunsaturated Fatty Acids (PUFAs)	Activation of peroxisome proliferator-activated receptor s (PPARs). Neuronal hyperpolarization by Ion channels modulation (e.g., activation of K2P channels and Na/K ATPase, inhibition of calcium and voltage-gated sodium channels). Increase in uncoupling proteins (UCP) and decrease of reactive oxygen species (ROS).
(3) Protection from apoptosis and cell death and inhibition of pro-apoptotic factors	Increase in calbindin. Inhibition of pore formation in mitochondria. Inhibition of pro-apoptotic factors (e.g., caspase 3).
(4) Change of intestinal microbiota	Possible increase in the seizure threshold caused by alteration of gut microbiome (e.g., elevation of some putative bacteria, such as Akkermansia muciniphila and Parabacteroides) in mouse models and human studies.
(5) Change of proinflammatory and anti-inflammatory mediators production	Reduced production of interleukin 1b and other cytokines in mouse models treated with ketogenic diet.

References: [8,28,29,30].

**Table 3 nutrients-12-02645-t003:** Most relevant articles on the use of KD (associated with AED therapy) in pediatric refractory epilepsy.

References	Type of Study	Participants		KD Ratio	Fasting	Outcomes						Follow-Up	Statistical Significance
[]												(Months)	
		Number	Age (Years)			KD Group			Control Group				
						Number	SFR >50%(*n*)	SFR >50%	Number	SFR >50%(*n*)	SFR >50%		
Vining EP, et al. 1998 [59]	Observational	51	8–1 years	Classic KD/4:1	Yes	45	54	28	/	/	/	3	*p* < 0.001
						35	53	27	/	/	/	6	NA
						24	40	20	/	/	/	12	NA
Freeman JM, et al. 1998 [60]	Observational	150	0.3–16 years	Classic KD/3–4:1	Yes	125	89	60	/	/	/	3	NA
						106	77	51	/	/	/	6	NA
						83	75	50	/	/	/	12	NA
Freitas A, et al. 2007 [61]	Observational	54	1–12 years	Classic KD/4:1	Yes	54	47	88.9	/	/	/	2	NA
						49	44	89.4	/	/	/	6	Efficacy for generalized epilepsy *p* < 0.001
						39	38	97.5	/	/	/	12	NA
						29	29	100	/	/	/	24	NA
Wu YJ, et al. 2016 [62]	Observational	87	0.5–16 years	Classic KD/4:1	Yes	87	47	41	/	/	/	1	NA
						87	51	44	/	/	/	3	NA
						87	52	45	/	/	/	6	Positive correlation between increased cognition and KD efficacy after 3 months (*p* = 0.003)
Neal EG, et al. 2008 [23]	RCT	145	2–16 years	Classic KD or MCT/2–5:1 or 40−60%	No	54	20	38	49	3	6	12	*p* < 0.0001
El-Rashidy OF, et al. 2013 [63]	RCT	10	26 ± 0.9 mon	Classic KD/4:1	NA	25	12	49.41	15	1	8.31	6	*p* < 0.005
Sharma S, et al. 2013 [64]	RCT	102	2–14 years	MAD (carbohydrate 10 g/die)	NA	50	26	52	52	6	11.5	3	*p* < 0.001
Sharma S, et al. 2016 [65]	RCT	81	2–14 years	MAD	NA	41	23	56.1	40	3	7.5	3	*p* < 0.0001
Lambrechts DAJE, et al. 2017 [66]	RCT	48	1–17 years	prevalent MCT	No	26	13	50	22	4	18.2	4	*p* = 0.024
Wijnen BFM, et al. 2017 [67]	RCT	48	1–18 years	Classic KD or MCT	NA	26	9	35	22	4	18	16	*p* = 0.171
Sourbron J, et al. 2020 [68]	Review-meta-analysis 5 RCTs	472						35–56.1			6–18.2		*p* < 0.001

Abbreviations: KD, ketogenic diet; AED, anti-epileptic drug; SFR, seizure frequency reduction; NA, not available; RCT, randomized controlled trial; MCT, medium-chain triglycerides; MAD, modified Atkins diet.

**Table 4 nutrients-12-02645-t004:** Most relevant articles on the use of KD (associated with AED therapy) in infantile spasms.

References	Type of Study	Participants		KD Ratio	Fasting	Outcomes			Follow-Up	Statistical Significance
[]									(Months)	
		Number	Age			Pt on KD	SFR >50%	SFR >50%		
			(Months)				(*n*)	(%)		
Than KD, et al. 2005 [69]	Retrospective observational	25	NA	3–4:1	Yes	25	NA	NA	NA	NA
Eun SH, et al. 2006 [70]	Retrospective observational	43	1–14	3–4:1	Yes	43	12	27,9	NA	NA
						35	30	85,7	3	NA
						25	23	92	6	NA
Kossoff EH, et al. 2008 [71]	Retrospective observational	13	Median (range): 5 (2–10)	3–4:1	Yes	13	8	62	1	*p* = 0.06
Hong AM, et al. 2010 [72]	Prospective observational	104	Median: 4.8	3–4:1	Yes	86	66	63	3	NA
						76	67	64	6	*p* = 0.84
					No	53	76	73	9	NA
						47	80	77	12	NA
						28	80	77	24	NA
Caraballo R, et al. 2011 [73]	Retrospective observational	12	NA	3–4:1	NA	12	9	75	NA	NA
Numis AL, et al. 2011 [74]	Retrospective observational	26	Mean ± SD: 19.5 ± 2.2	3–4:1	No	26	12	46.2	3	*p* = 0.02
						21	11	52.4	6	*p* = 0.02
						19	12	63.2	12	*p* = 0.02
Lee J, et al. 2013 [75]	Retrospective observational	14	NA	NA	NA	14	11	78,6	NA	NA
Li B, et al. 2013 [76]	Prospective observational	31	7–84	4:01	Yes	31	21	67,74	1	NA
							22	70,97	3	NA
Pires ME, et al. 2013 [77]	Prospective observational	17	Mean ± SD: 9.4 ± 1.1	3–4:1	No	17	15	88.2	3	NA
						16	14	87.5	6	NA
Kayyali HR, et al. 2014 [78]	Prospective observational	20	4–35	3–3.5:1	No	20	14	70	3	NA
						20	13	72.2	6	NA
						17	13	76.5	12	NA
						8	8	100	24	NA
Hirano Y, et al. 2015 [79]	Retrospective observational	6	9–40	3–4:1	NA	6	5	83,3	3	NA
Ville D, et al. 2015 [80]	Retrospective observational	23	NA	4:01	Yes	23	4	17,4	NA	NA
Hussain SA, et al. 2016 [81]	Retrospective observational	22	12.5–38.7	3–4:1	No	22	7	35	3	NS
						20	6	35.3	6	NS
						17	2	20	12	NS

Abbreviations: KD, ketogenic diet; AED, anti-epileptic drug; SFR, seizure frequency reduction; NA, not available; NS, not significant; SD, standard deviation.

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
