# Peer review of "Diet in the Treatment of Epilepsy: What We Know So Far"

_nutrients, 2020, doi:10.3390/nu12092645_

Round 1

Reviewer 1 Report

I have read a review entitled “Diet in the treatment of epilepsy: what we know so far” that describes the use of ketogenic diet in the treatment of drug-resistant epilepsy, with particular attention to different forms of epileptic disorders in children. The authors have characterized the ketogenic diet taking into account its modifications (i.e., modified Atkins diet, low-glycemic index treatment, medium-chain triglyceride diet), possibilities of its application in specific forms of epilepsy disorders and potential mechanisms of anticonvulsant action. Much attention has been paid to the analysis of the results of previous studies showing the effectiveness of using ketogenic diet. Unfortunately, this analysis focus on cases of pediatric epilepsy, the results of using the ketogenic diet in adult patients with epilepsy are poorly described. In my opinion, the work is written well, with correct English language, only slight punctuation errors are present. The review is logical and coherent. I think that manuscript could be accepted for publication in the present form.

Reviewer 2 Report

In the current article, the authors reviewed on the possible clinical efficacy of different dietary regimens, underlying pathophysiological mechanisms, safety and tolerability both in pediatric and adult epilepsy.

There are a few major concerns in the manuscript as follows.

Major Concerns

  1. The authors mentioned that the review article is focused both on the childhood and adult epilepsy. However, not much information or potential pathophysiological mechanisms have been discussed. Please provide more details, also in a tabular form similar to the one for the pediatric epilepsy.
  2. Also, give more details, may be in a table, the differences between the dietary regimens etc., between these groups.
  3. It may be a good idea to also provide figures such as a figure with mechanisms.

Reviewer 3 Report

  • Please provide reference for line 1 if the same sentence is not repeated in introduction again.
  • Please keep ‘classic KD’ instead of classical KD all over the article. Please avoid using abbreviations in title, headings or subheadings.
  • Line 60-62: Diets: instead of indicating what should be done, please state what usually done.
  • Instead of ‘fat/protein’ in a sentence please use fat per protein, all over article.
  • Line 74: please keep original hospital name e.g. Massachusetts General Hospital.
  • Line 90: change name to ‘physiology or pathophysiology of Ketogenic diet and epilepsy’
  • Line 94: instead of ‘thanks to’ please use ‘because of’.
  • Line 96: Acetoacetate and beta hydroxy butyrate
  • Line 99: replace ‘thanks’ with ‘by’
  • Line 109: instead of α-ketoglutarate in GABA, please change to α-ketoglutarate which is a precursor of GABA
  • Line 104-122, please state all inhibitory neurotransmitters of brain at first in a sentence and then please describe the mechanisms of each of those chemicals and how they are modified by KD.
  • Line 153: please use species instead of spp.
  • Line 156: please use IL1b to interleukin1b
  • 162: PPARs expand the abbreviation since it was mentioned first time, which is different from PPARγ
  • Table 1: please keep articles in chronological order according to year in observational studies.
  • Line 172: please mention the years range of articles in table 1.
  • The table below table 1 appears to be separate, but with no heading to it. It is better to name it as table 2, please state what it shows.
  • 1st table has ‘observational’ word and the 2nd table below it has prospective/retrospective words. Please use these descriptive words uniformly. E.g. observational/cohort/prospective or observational/(cohort or case control)/retrospective.
  • Line 242-284: it is a good idea to define syndromes in a sentence before stating the advantages of the KD diet in those syndromes.
  • In subheadings like ‘general AEs’, rather than abbreviations, it is better to expand AEs to adverse events. Please avoid abbreviations in subheadings or headings in general for whole article.
  • Line 357-369: Antiepileptic medications, obesity (nutrition status), and ketogenic diet interactions may have important practical implications on sustainability, safety and efficacy of KD in epilepsy (due to current rise of pediatric obesity). Please comment. If you find evidence in literature please provide reference.
  • Line 402-403, nutrition status of epileptic patient, responsiveness of epilepsy to other treatment modalities might also be important in making a decision.
  • Dietary modification as a treatment modality produces complex interactions between the disease, patient and the diet. In your conclusion, please comment on if KD could be used as single treatment or adjunct to other treatment modalities, 1st line of treatment or added later etc. If you find evidence, please provide reference.
  • Please comment about KD role in epilepsy, about current gaps and how they can be filled and on future research direction in your conclusion.

Reviewer 4 Report

Please check the comments from the attached article. This article can focus more on the efficacy of KD for the specific types of seizures that have not been investigated much. (e.g., infantile syndrome)

Round 2

Reviewer 3 Report

The revised manuscript looks more organized. But needs extensive English language editing.

E.g. line 255: it should be ‘growth’ instead of ‘grow’;

line 257: ‘lipid profile’ instead of ‘lipidic profile’.

These kind of mistakes are minor, but with proper editing, avoidable.

Current gaps or limitations in KD research, and how they can be filled in future research should be improved. Please make it more organised (e.g most important gap to least important gap as per your opinion) in discussion as well as briefly in conclusion.
